# Input margins can predict generalization too

## Abstract

Understanding generalization in deep neural networks is an active area of research. A promising avenue of exploration has been that of margin measurements: the shortest distance to the decision boundary for a given sample or its representation internal to the network. While margins have been shown to be correlated with the generalization ability of a model when measured at its hidden representations (hidden margins), no such link between large margins and generalization has been established for *input margins*. We show that while input margins are not generally predictive of generalization, they can be if the search space is appropriately constrained. We develop such a measure based on input margins, which we refer to as 'constrained margins'. The predictive power of this new measure is demonstrated on the 'Predicting Generalization in Deep Learning' (PGDL) dataset and contrasted with hidden representation margins. We find that constrained margins achieve highly competitive scores and outperform other margin measurements in general.

## 1 Introduction

Our understanding of the generalization ability of deep neural networks (DNNs) remains incomplete. Various bounds on the generalization error for classical machine learning models have been proposed based on the complexity of the hypothesis space [1, 2]. However, this approach paints an unfinished picture when considering modern DNNs [3]. Generalization in DNNs is an active field of study and updated bounds are proposed on an ongoing basis [4, 5, 6, 7].

A complementary approach to developing theoretical bounds is to develop empirical techniques that are able to predict the generalization ability of certain families of DNN models. The 'Predicting Generalization in Deep Learning' (PGDL) challenge, exemplifies such an approach. The challenge was held at NeurIPS 2020 [8] and provides a useful test bed for evaluating *complexity measures*, where a complexity measure is a scalar-valued function that relates a model's training data and parameters to its expected performance on unseen data. Such a predictive complexity measure would not only be practically useful but could lead to new insights into how DNNs generalize.

In this work, we focus on classification margins in deep neural classifiers. It is important to note that the term 'margin' is, often confusingly, used to refer to 1) output margins [9], 2) input margins [10], and 3) hidden margins [11], interchangeably. Here (1) is a measure of the difference in class output values, while (2) or (3) is concerned with measuring the distance from a sample to its nearest decision boundary in either input or hidden representation space, respectively. In this work, we focus on input and hidden margins.

While margins measured at the hidden representations of deep neural classifiers have been shown to be predictive of a model's generalization, this link has not been established for input space margins. We show that, in several circumstances, the classical definition of input margin does *not* predict generalization, but a direction-constrained version of this metric does: a quantity we refer to as

*constrained margins*. By measuring margins in directions of 'high utility', that is, directions that are expected to be more useful to the classification task, we are able to better capture the generalization ability of a trained DNN.

We make several contributions:

1. Demonstrate the first link between large input margins and generalisation performance, by developing a new input margin-based complexity measure that achieves highly competitive performance on the PGDL benchmark and outperforms several contemporary complexity measures.

2. Show that margins do not necessarily need to be measured at multiple hidden layers to be predictive of generalization, as suggested in [11].

3. Provide a new perspective on margin analysis and how it applies to DNNs, that of finding high utility directions along which to measure the distance to the boundary instead of focusing on finding the shortest distance.

## 2 Background

This section provides an overview of existing work on 1) measuring classification margins and their relationship to generalization, and 2) the PGDL challenge and related complexity measures.

### 2.1 Classification Margins and Generalization

Considerable prior work exists on understanding classification margins in machine learning models [12, 13]. The relation between margin and generalization is well understood for classifiers such as support vector machines (SVMs) under statistical learning theory [1]. However, the non-linearity and high dimensionality of DNN decision boundaries complicate such analyses, and precisely measuring these margins is considered intractable [14, 15].

A popular technique (which we revisit in this work) is to approximate the classification margin using a first-order Taylor approximation. Elsayed et al. [16] use this method in both the input and hidden space, and then formulate a loss function that maximizes these margins. However, while this results in a measurable increase in margin, it does not result in any significant gains in test accuracy. In a seminal paper, Jiang et al. [11] utilize the same approximation in order to predict the generalization gap of a set of trained networks by training a linear regression model on a summary of their hidden margin distributions. Natekar and Sharma [17] demonstrate that this measure can be further improved if margins are measured using the representations of Mixup [18] or augmented training samples. Similarly, Chuang et al. [6] introduce novel generalization bounds and slightly improve on this metric by proposing an alternative cluster-aware normalization scheme ($k$-variance [19]).

Input margins are generally considered from the point of view of adversarial robustness, and many techniques have been developed to generate adversarial samples on or near the decision boundary. Examples include: the Carlini and Wagner Attack [20], Projected Gradient Descent [21], and DeepFool [22]. Some of these studies have investigated the link between adversarial robustness and generalization, often concluding that an inherent trade-off exists [23, 24, 25]. However, this conclusion and its intricacies are still being debated [26].

Yousefzadeh and O'Leary [14] formulate finding a point on the decision boundary as a constrained minimization problem, which is solved using an off-the-shelf optimization method. While this method is more precise, it comes at a great computational cost. To alleviate this, dimensionality reduction techniques are used in the case of image data to reduce the number of input features. In this case, the classification margin is used for the purpose of model interpretability.

In this work we propose a modification to the Taylor approximation of the input classification margin (and its iterative alternative DeepFool) in order for it to be more predictive of generalization.

### 2.2 Predicting Generalization in Deep Learning

The objective of this challenge was to design a complexity measure to rank models according to their generalization gap. More precisely, participants only had access to a set of trained models, along with

their parameters and training data, and were tasked with ranking the models within each set according to their generalization gap. Each solution was then evaluated on how well its ranking aligns with the true ranking on a held-out set of tasks, which was unknown to the competitors.

In total, there are 550 trained models across 8 different tasks and 6 different image classification datasets, where each task refers to a set of models trained on the same dataset with varying hyperparameters and subsequent test accuracy. Tasks 1, 2, 4, and 5 were available for prototyping and tuning complexity measures, while Task 6 to 9 were used as a held-out set. There is no task 3. The final average score on the test set was the only metric used to rank the competitors. Conditional mutual information (CMI) is used as evaluation metric, which measures the conditional mutual information between the complexity measure and true generalization gap, given that a set of hyperparameter types are observed. This is done in order to prevent spurious correlations resulting from specific hyperparameters, a step towards establishing whether a causal relationship exists.

All models were trained to approximately the same, near zero, training loss. Note that this implies that ranking models according to either their generalization gap or test accuracy is essentially equivalent.

Several interesting solutions were developed during the challenge: In addition to the modification of hidden margins mentioned earlier, the winning team [17] developed several prediction methods based on the internal representations of each model. Their best-performing method measures clustering characteristics of hidden layers (using Davies-Bouldin Index [27]), and combines this with the model's accuracy on Mixup-augmented training samples. In a similar fashion, the runners-up based their metrics on measuring the robustness of trained networks to augmentations of their training data [28].

After the competition's completion, the dataset was made publicly available, inspiring further research: Schiff et al. [29] generated perturbation response curves that 'capture the accuracy change of a given network as a function of varying levels of training sample perturbation' and develop statistical measures from these curves. They produced eleven complexity measures with different types of sample Mixup and statistical metrics.

While several of the methods rely on using synthetic samples (e.g. Mixup), Zhang et al. [30] take this to the extreme and generate an artificial test set using pretrained generative adversarial networks (GANs). They demonstrate that simply measuring the classification accuracy on this synthetic test set is very predictive of a model's generalization. While practically useful, this method does not make a link between any characteristics of the model and its generalization ability.

## 3 Theoretical approach

This section provides a theoretical overview of the proposed complexity measure. We first explain our intuition surrounding classification margins, before mathematically formulating constrained margins.

### 3.1 Intuition

A correctly classified training sample with a large margin can have more varied feature values, potentially due to noise, and still be correctly classified. However, as we will show, input margins are not generally predictive of generalization. This observation is supported by literature regarding adversarial robustness, where it has been shown that adversarial retraining (which increases input margins) can negatively affect generalization [23, 25].

Stutz et al. [26] provide a plausible reason for this counter-intuitive observation: Through the use of Variational Autoencoder GANs they show that the majority of adversarial samples leave the class-specific data manifold of the samples' class. They offer the intuitive example of black border pixels in the case of MNIST images, which are zero for all training samples. Samples found on the decision boundary which manipulate these border pixels have a zero probability under the data distribution, and they do not lie on the underlying manifold.

We leverage this intuition and argue that any input margin measure that relates to generalization should measure distances along directions that do not rely on spurious features in the input space. The intuition is that, while nearby decision boundaries exist for virtually any given training sample, these nearby decision boundaries are likely in directions which are not inherently useful for test set classification, i.e. they diverge from the underlying data manifold.

More specifically, we argue that margins should be measured in directions of 'high utility', that is, directions that are expected to be useful for characterising a given dataset, while ignoring those of lower utility. In our case, we approximate these directions by defining high utility directions as directions which explain a large amount of variance in the data. We extract these using Principal Component Analysis (PCA). While typically used as a dimensionality reduction technique, PCA can be interpreted as learning a low-dimensional manifold [31], albeit a linear one. In this way, the PCA manifold identifies subspaces that are thought to contain the variables that are truly relevant to the underlying data distribution, which the out-of-sample data is assumed to also be generated from. In the following section, we formalize such a measure.

## 3.2 Constrained Margins

We first formulate the classical definition of an input margin [14], before adapting it for our purpose. Let $f : X \to \mathbb{R}^{|N|}$ denote a classification model with a set of output classes $N = \{1 \dots n\}$, and $f_k(\mathbf{x})$ the output value of the model for input sample $\mathbf{x}$ and output class $k$. For a correctly classified input sample $\mathbf{x}$, the goal is to find the closest point $\hat{\mathbf{x}}$ on the decision boundary between the true class $i$ (where $i = \arg\max_k(f_k(\mathbf{x}))$) and another class $j \neq i$. Formally, $\hat{\mathbf{x}}$ is found by solving the constrained minimization problem:

$$\underset{\hat{\mathbf{x}} \in [L,U]}{\arg\min} ||\mathbf{x} - \hat{\mathbf{x}}||_2 \tag{1}$$

with $L$ and $U$ the lower and upper bounds of the search space, respectively, such that

$$f_i(\hat{\mathbf{x}}) = f_j(\hat{\mathbf{x}}) \tag{2}$$

for $i$ and $j$ as above.

The margin is then given by the Euclidean distance between the input sample, $\mathbf{x}$, and its corresponding sample on the decision boundary, $\hat{\mathbf{x}}$. We now adapt this definition in order to define a 'constrained margin'. Let the set $P = \{\mathbf{p}_1, \mathbf{p}_2, ..., \mathbf{p}_m\}$ denote the first $m$ principal component vectors of the training dataset, that is, the $m$ orthogonal principal components which explain the most variance. Such principal components are straightforward to extract by first standardizing (z normalizing) each feature individually, and then calculating the eigenvectors of the covariance matrix of the standardized training data.

We now restrict $\hat{\mathbf{x}}$ to any point consisting of the original sample $\mathbf{x}$ plus a linear combination of these principal component vectors, that is, for some coefficient vector $\mathbf{B} = [\beta_1, \beta_2, ..., \beta_m]$

$$\hat{\mathbf{x}} \triangleq \mathbf{x} + \sum_{i=1}^{m} \beta_i \mathbf{p}_i \tag{3}$$

Substituting $\hat{\mathbf{x}}$ into the original objective function of Equation (1), the new objective becomes

$$\min_{\beta} || \sum_{i=1}^{m} \beta_i \mathbf{p}_i ||_2 \tag{4}$$

such that Equation (2) is approximated within a certain tolerance. For this definition of margin, the search space is constrained to a lower-dimensional subspace spanned by the principal components with point $\mathbf{x}$ as origin, and the optimization problem then simplifies to finding a point on the decision boundary within this subspace. By doing so, we ensure that boundary samples that rely on spurious features (that is, in directions of low utility) are not considered viable solutions to Equation (1). Note that this formulation does not take any class labels into account for identifying high utility directions.

While it is possible to solve the constrained minimization problem using a constrained optimizer [14], we approximate the solution by adapting the previously mentioned first-order Taylor approximation [16, 32], which greatly reduces the computational cost. The Taylor approximation of the constrained margin $d(\mathbf{x})$ for a sample $\mathbf{x}$ between classes $i$ and $j$ when using an $L2$ norm is given by

$$d(\mathbf{x}) = \frac{f_i(\mathbf{x}) - f_j(\mathbf{x})}{|| [ \nabla_{\mathbf{x}} f_i(\mathbf{x}) - \nabla_{\mathbf{x}} f_j(\mathbf{x}) ] \mathbf{P}^T ||_2} \tag{5}$$

where $\mathbf{P}$ is the $m \times n$ matrix formed by the top $m$ principal components with $n$ input features. The derivation of Equation (5) is included in the supplementary material.

The value $d(\mathbf{x})$ only approximates the margin and the associated discrepancy in Equation (2) can be large. In order to reduce this to within a reasonable tolerance, we apply Equation (5) in an iterative manner, using a modification of the well-known DeepFool algorithm [22]. DeepFool was defined in the context of generating adversarial samples with the smallest possible perturbation, which is in effect very similar to finding the nearest point on the decision boundary with the smallest violation of Equation (2).

To extract the DeepFool constrained margin for some sample $\mathbf{x}$, the Taylor approximation of the constrained margin is calculated between the true class $i$ and all other classes $j$, individually. A small step (scaled by a set learning rate) is then taken in the lower-dimensional subspace in the direction corresponding to the class with smallest margin. This point is then transformed back to the original feature space and the process is repeated until the distance changes less than a given tolerance in comparison to the previous iteration. The exact process to calculate a DeepFool constrained margin is described in Algorithm 1. Note that we also clip $\hat{\mathbf{x}}$ according to the minimum and maximum feature values of the dataset after each step, which ensures that the point stays within the bound constraints expressed in Equation 1. While this is likely superfluous when generating normal adversarial samples – they are generally very close to the original $\mathbf{x}$ – it is a consideration when the search space is constrained, with clipped margins performing better. (See the supplementary material for an ablation analysis of clipping.)

---

**Algorithm 1** DeepFool constrained margin calculation

---

**Input**: Sample $\mathbf{x}$, classifier $f$, principal components $\mathbf{P}$
**Parameter**: Stopping tolerance $\delta$, Learning rate $\gamma$, Maximum iterations $max$
**Output**: Distance $d_{best}$, Equality violation $v_{best}$

---

1: $\hat{\mathbf{x}} \leftarrow \mathbf{x}, i \leftarrow \arg\max f_k(\mathbf{x}), d \leftarrow 0, v_{best} \leftarrow \infty, c \leftarrow 0$
2: **while** $c \leq max$ **do**
3:     **for** $j \neq i$ **do**
4:         $o_j \leftarrow f_i(\hat{\mathbf{x}}) - f_j(\hat{\mathbf{x}})$
5:         $\mathbf{w}_j \leftarrow [\nabla f_i(\hat{\mathbf{x}}) - \nabla f_j(\hat{\mathbf{x}})]\mathbf{P}^T$
6:     **end for**
7:     $l \leftarrow \arg\min_{j \neq i} \frac{|o_j|}{||\mathbf{w_j}||_2}$
8:     $\mathbf{r} \leftarrow \frac{|o_l|}{||\mathbf{w}_l||_2^2}\mathbf{w}_l\mathbf{P}$
9:     $\hat{\mathbf{x}} \leftarrow \hat{\mathbf{x}} + \gamma\mathbf{r}$
10:     $\hat{\mathbf{x}} \leftarrow \text{clip}(\hat{\mathbf{x}})$
11:     $v \leftarrow |o_l|$
12:     $d \leftarrow ||\mathbf{x} - \hat{\mathbf{x}}||_2$
13:     **if** $v \geq v_{best}$ **or** $|d - d_{best}| < \delta$ **then**
14:         **return** $d_{best}, v_{best}$
15:     **else**
16:         $v_{best} \leftarrow v$
17:         $d_{best} \leftarrow d$
18:         $c \leftarrow c + 1$
19:     **end if**
20: **end while**
21: **return** $d_{best}, v_{best}$

---

# 4 Results

We investigate the extent to which constrained margins are predictive of generalization by comparing the new method with current alternatives. In Section 4.1 we describe our experimental setup. Following this, we do a careful comparison between our metric and existing techniques based on standard input and hidden margins (Section 4.2) and, finally, we compare with other complexity measures (Section 4.3).

## 4.1 Experimental setup

For all margin-based measures our indicator of generalization (complexity measure) is the mean margin over $5\,000$ randomly selected training samples, or alternatively the maximum number available for tasks with less than $5\,000$ training samples. Only correctly classified samples are considered, and the same training samples are used for all models of the same task. To compare constrained margins to input and hidden margins we rank the model test accuracies according to the resulting indicator and calculate the Kendall's rank correlation [33], as used in [34]. This allows for a more interpretable comparison than CMI. (As CMI is used throughout the PGDL challenge, we also include the resulting CMI scores in the supplementary material.) To compare constrained margins to published results of other complexity measures, we measure CMI between the complexity measure and generalization gap and contrast this with the reported scores of other methods.

As a baseline we calculate the **standard input margins** ('Input') using the first order Taylor approximation (Equation 5 without the subspace transformation), as we find that it achieves better results than the iterative DeepFool variant and is therefore the stronger baseline; see the supplementary material for a full comparison.

Creating a complexity measure from **hidden margins** ('Hidden') raises the question of which hidden layers to consider. Jiang et al. [11] consider three equally spaced layers, Natekar and Sharma [17] consider all layers, and Chuang et al. [6] consider either the first or last layer only. We calculate the mean hidden margin (using the Taylor approximation) for all these variations and find that for the tasks studied here, using the first layer performs best, while the mean over all layers comes in second. We include both results here. (A full analysis is included in the supplementary material.) We normalize each layer's margin distribution by following [11], and divide each margin by the total feature variance at that layer.

Our **constrained margin** complexity measure ('Constrained') is obtained using Algorithm 1, although in practice we implement this in a batched manner. Empirically, we find that the technique is not very sensitive with regard to the selection of hyperparameters and a single learning rate ($\gamma = 0.25$), tolerance ($\delta = 0.01$), and max iterations ($max = 100$) is used across all experiments. The number of principal components for each dataset is selected by plotting the explained variance (of the train data) per principal component in decreasing order on a logarithmic scale and applying the elbow method using the Kneedle algorithm from Satopaa et al [35]. This results in a very low-dimensional search space, ranging from $3$ to $8$ principal components for the seven unique datasets considered.

In order to prevent biasing our metric to the PGDL test set (tasks 6 to 9) we did not perform any tuning or development of the complexity measure using these tasks, nor do we tune any hyperparameters per task. The choice of principal component selection algorithm was done after a careful analysis of Tasks 1 to 5 only, see additional details in the supplementary material. In terms of computational expense, we find that calculating the entire constrained margin distribution only takes 1 to 2 minutes per model on an Nvidia A30.

## 4.2 Margin complexity measures

In Table 1 we show the Kendall's rank correlation obtained when ranking models according to constrained margin, standard input margins, and hidden margins. It can be observed that standard input margins are not predictive of generalization for most tasks and, in fact, show a negative correlation for some. This unstable behaviour is supported by ongoing work surrounding adversarial robustness and generalization [23, 24, 25]. Furthermore, we observe a very large performance gap between constrained and standard input margins, and an increase from $0.24$ to $0.66$ average rank correlation is observed by constraining the margin search. This strongly supports our initial intuitions.

In the case of hidden margins, performance is more competitive, however, constrained margins still outperform hidden margins on $6$ out of $8$ tasks. One also observes that the selection of hidden layers can have a very large effect, and the discrepancy between the two hidden-layer selections is significant. Given that our constrained margin measurement is limited to the input space, there are several advantages: 1) no normalization is required, as all models share the same input space, and 2) the method is more robust when comparing models with varying topology, as no specific layers need to be selected.

Table 1: Kendall's rank correlation between mean margin and test accuracy for constrained, standard input, and hidden margins using the first or all layer(s). Models in Task 4 are trained with batch normalization while models in Task 5 are trained without. There is no Task 3.

| Task | Architecture | Dataset | Constrained | Input | Hidden (1st) | Hidden (all) |
|---|---|---|---|---|---|---|
| 1 | VGG | CIFAR10 | **0.8040** | 0.0265 | 0.5794 | 0.7825 |
| 2 | NiN | SVHN | **0.8672** | 0.6841 | 0.7037 | 0.8281 |
| 4 | FCN | CINIC10 | 0.6651 | 0.6251 | **0.7958** | 0.2707 |
| 5 | FCN | CINIC10 | 0.2292 | 0.3571 | **0.5427** | 0.1329 |
| 6 | NiN | OxFlowers | **0.8008** | -0.1351 | 0.4427 | 0.2839 |
| 7 | NiN | OxPets | **0.5027** | 0.3215 | 0.3623 | 0.3481 |
| 8 | VGG | FMNIST | **0.6004** | -0.1233 | -0.0656 | 0.1859 |
| 9 | NiN | CIFAR10 (augmented) | **0.8145** | 0.1573 | 0.7097 | 0.4556 |
| Average | | | **0.6605** | 0.2392 | 0.5088 | 0.4110 |

## 4.3 Other complexity measures

To further assess the predictive power of constrained margins, we compare our method to the reported CMI scores of several other complexity measures. We compare against three solutions from the winning team [17], as well as the best solutions from two more recent works [6, 29], where that of Schiff et al. [29] has the highest average test set performance we are aware of. We do not compare against pretrained GANs [30]. The original naming of each method is kept. Of particular relevance are the $MM$ and $AM$ columns, which are hidden margins applied to Mixup and Augmented samples, as well as $k$V-Margin and $k$V-GN-Margin which are output and hidden margins with $k$-Variance normalization, respectively. The results of this comparison are shown in Table 2.

One observes that constrained margins achieve highly competitive scores, and in fact, outperform all other measures on 4 out of 8 tasks. It is also important to note that the *MM* and *AM* columns show that hidden margins can be improved in some cases if they are measured using the representations of Mixup or augmented training samples. That said, these methods still underperform on average in comparison to constrained input margins, which do not rely on any form of data augmentation.

Table 2: Conditional Mutual Information (CMI) scores for several complexity measures on the PGDL dataset. Acronyms: $DBI$=Davies Bouldin Index, $LWM$=Label-wise Mixup, $MM$=Mixup Margins, $AM$=Augmented Margins, $kV$=$k$-Variance, $GN$=Gradient Normalized, $Gi$=Gini coefficient, $Mi$=Mixup. Test set average is the average over tasks 6 to 9. There is no Task 3. †Indicates a margin-based measure.

| Task | Natekar and Sharma | | | Chuang et al. | | Schiff et al. | Ours |
|---|---|---|---|---|---|---|---|
| | DBI*LWM | MM† | AM† | $k$V-Margin 1st† | $k$V-GN-Margin 1st† | PCA Gi&Mi | Constrained Margin† |
| 1 | 00.00 | 01.11 | 05.73 | 05.34 | 17.95 | 0.04 | **39.37** |
| 2 | 32.05 | 47.33 | 44.60 | 26.78 | 44.57 | 38.08 | **51.12** |
| 4 | 31.79 | 43.22 | **47.22** | 37.00 | 30.61 | 33.76 | 21.48 |
| 5 | 15.92 | **34.57** | 22.82 | 16.93 | 16.02 | 20.33 | 05.12 |
| 6 | 43.99 | 11.46 | 08.67 | 06.26 | 04.48 | **40.06** | 30.52 |
| 7 | 12.59 | **21.98** | 11.97 | 02.07 | 03.92 | 13.19 | 12.60 |
| 8 | 09.24 | 01.48 | 01.28 | 01.82 | 00.61 | 10.30 | **13.54** |
| 9 | 25.86 | 20.78 | 15.25 | 15.75 | 21.20 | 33.16 | **51.46** |
| Test set average | 22.92 | 13.93 | 09.29 | 06.48 | 07.55 | 23.62 | **27.03** |

## 5 A closer look

In this section we do a further analysis of constrained margins. In Section 5.1 we investigate how the performance of constrained margins changes when lower utility subspaces are considered, whereafter we discuss limitations of the method in Section 5.2.

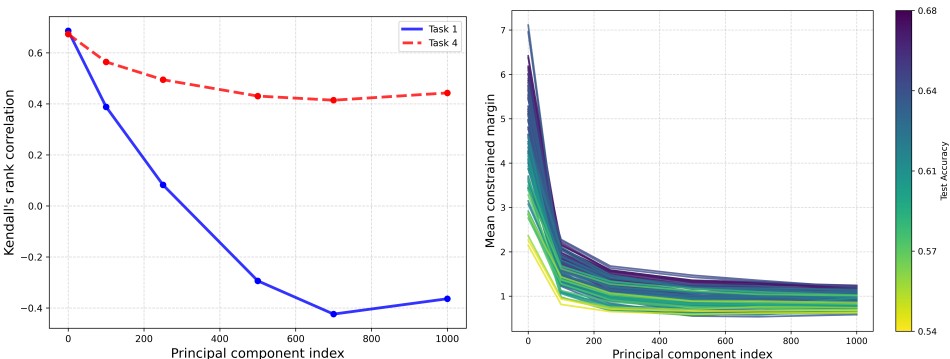

Figure 1: Comparison of high to low utility directions using subspaces spanned by 10 principal components, x-axis indicates the first component in each set of principal components. Left: Kendall's rank correlation for Task 1 (blue solid line) and 4 (red dashed line). Right: Mean constrained margin for models from Task 4.

## 5.1 High to low utility

We examine how high utility directions compare to those of lower utility when calculating constrained margins. This allows us to further test our approach, as one would expect that margins measured using the lower-ranked principal components should be less predictive of a model's performance.

We calculate the mean constrained margin using select subsets of 10 contiguous principal components in descending order of explained variance. For example, we calculate the constrained margins using components 1 to 10, then 100 to 109, etc. This allows us to calculate the distance to the decision boundary using 10 dimensional subspaces of decreasing utility. We, once again, make use of 5 000 training samples. We restrict ourselves to analysing the training set of tasks (tasks 1-5) and consider one task where constrained margins perform very well (Task 1) and one with poorer performance (Task 4). Figure 1 (left) shows the resulting Kendall's rank correlation for each subset of principal components indexed by the first component in each set (principal component index). The right-hand side shows the mean margin of all models from Task 4 at each subset.

As expected, the first principal components lead to margins that are more predictive of generalization. We see a gradual decrease in predictive power when considering later principal components. Task 1 especially suffers this phenomenon, reaching negative correlations. This supports the idea that utilizing the directions of highest utility is a necessary aspect of input margin measurements. Additionally, one observes that the mean margin also rapidly decreases after the first few sets of principal components. After the point shown here (index 1 000), we find that the mean margin increases as DeepFool struggles to find samples on the decision boundary within the bound constraints. Due to this, it is difficult to draw any conclusions from an investigation of the lower-ranked principal components. This also points to the notion that the adversarial vulnerability of modern DNNs is in part due to nearby decision boundaries in the directions of the mid-tier principal components (the range of 100 to 1 000).

## 5.2 Limitations

It has been demonstrated that our proposed metric performs well and aligns with our intitial intuition. However, there are also certain limitations that require explanation. Empirically we observe that, for tasks where constrained margins perform well, they do so across all hyperparameter variations, with the exception of depth. This is illustrated in Figure 2 (left), which shows the mean constrained margin versus test accuracy for Task 1. We observe that sets of networks with two and six convolutional layers, respectively, each exhibit a separate relationship between margin and test accuracy. This discrepancy is not always as strongly present: for Task 6 all three depth configurations show a more similar relationship, as observed on the right of Figure 2, although the discrepancy is still present. The same trend holds for all tasks where it is observed (1, 2, 4, 6, 9). It appears that shallower networks model the input space in a distinctly different fashion than their deeper counterparts.

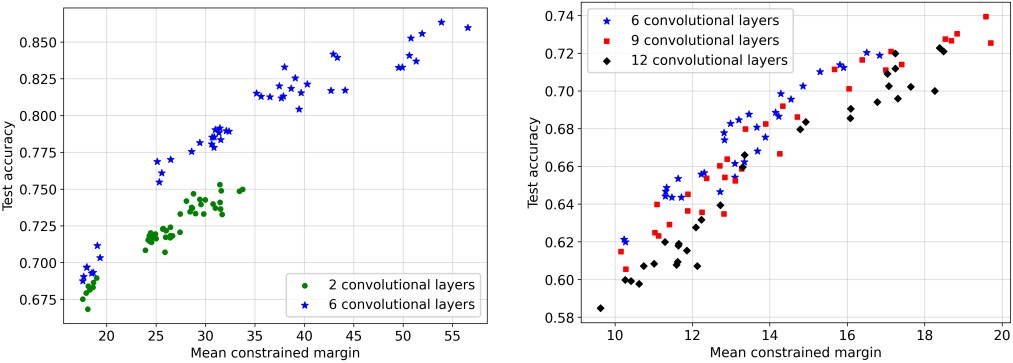

Figure 2: Mean constrained margin versus test accuracy for PGDL Task 1 (left) and 6 (right). Left: Models with 2 (green circle) and 6 (blue star) convolutional layers. Right: Models with 6 (blue star), 9 (red square), and 12 (black diamond) convolutional layers.

For tasks such as 5 and 7, where constrained margins perform more poorly, there is no single hyperparameter that appears to be the culprit. We do note that the resulting scatter plots of margin versus test accuracy never show points in the lower right (large margin but low generalization) or upper left (small margin but high generalization) quadrants. It is therefore possible that a larger constrained margin is always beneficial to a model's generalization, even though it is not always fully descriptive of its performance. Finally, while our approach to selecting the number of principal components is experimentally sound, the results can be further improved if the optimal number is known, see the supplementary material for details.

## 6   Conclusion

We have shown that constraining input margins to high utility subspaces can significantly improve their predictive power i.t.o generalization. Specifically, we have used the principal components of the data as a proxy for identifying these subspaces, which can be considered a rough approximation of the underlying data manifold.

Constraining the search to a warped subspace and using Euclidean distance to measure closeness is equivalent to defining a new distance metric on the original space. We are therefore, in effect, seeking a relevant distance metric to measure the closeness of the decision boundary. Understanding the requirements for such a metric remains an open question. Unfortunately, current approximations and methods for finding points on the decision boundary are largely confined to $L_p$ metrics. The positive results achieved with the current PCA-and-Euclidean-based approach provide strong motivation that this is a useful avenue to pursue. Furthermore, we believe that constrained margins can be used as a tool to further probe generalization, similar to the large amount of work that has been done surrounding standard input margins and characterization of decision boundaries.

In conclusion, we propose constraining input margins to make them more predictive of generalization in DNNs. It has been demonstrated that this greatly increases the predictive power of input margins, and also outperforms hidden margins and several other contemporary methods on the PGDL tasks. This method has the benefits of requiring no per-layer normalization, no arbitrary selection of hidden layers, and does not rely on any form of surrogate test set (e.g. data augmentation or synthetic samples).

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
