# OpenReview forum: "Input margins can predict generalization too"
_NeurIPS.cc/2023/Conference — Submitted to NeurIPS 2023_

### Official Review · Reviewer_Uvn7 · 2023-06-28

**Soundness:** 3 good
**Presentation:** 3 good
**Contribution:** 1 poor
**Rating:** 4
**Confidence:** 4

**Summary:**

The paper presents a modification to existing optimisation-based margin computation technique in neural networks to be used as an empirical complexity measure. The measure is evaluated on the Predicting Generalization in Deep Learning (PGDL) 2020 competition dataset (now all public) and shows better performance that other margin-based as well as the competition-winning method.

**Strengths:**

The introduced constraint to the margin definition is straight forward - restrict the walk towards the margin point from x to be within the top-m PCA components subspace of the training data.

The proposed modification seems to be effective according to the PGDL evaluation.


**Weaknesses:**

The new margin definition is not motivated theoretically. It seems like something sensible to do, but I am not convinced it gives us all that much insight beyond improving PGDL score. I suspect the approach might also fail on some datasets (as evident from the performance of the proposed method on Task 5 of Table 2 (though to authors credit, the do not shy away from talking about it in the Limitations section)). It doesn't seem like a rigorous derivation of a new principle, but a bit of a hack to improve an existing approach. I presume the point of the competition was to spur new ideas and new understanding in this domain. The proposed modification seem like a minor change, and other than improving the PGDL score, I don't find any new insight in terms of learning theory. As such, this seems like an exercise in optimisation for PGDL score, and I am not sure how the proposed PCA trick relates to gaining any new understanding about generalisation principles of neural networks.

**Questions:**

The assurance that no tuning was done on the (now public) PGDL test set is noble... but in the end, the test data is now public, and while the tuning of the hyperparameters might have beeen done on training/validation data, how often was the performance checked against test data? Technically, if task 6 to 9 was kept aside as a true test set, then it should have be evaluated once and only once at the end of all the simulations. Was that the case?

**Limitations:**

Yes, the authors speak about the limitations of their approach.

---

> ### Author Rebuttal · Authors · 2023-08-09
>
> We thank you for your review and comments. We shall first address your question, before providing a general rebuttal of the weaknesses you have pointed out.
>
> >"The assurance that no tuning was done on the (now public) PGDL test set is noble... but in the end, the test data is now public, and while the tuning of the hyperparameters might have beeen done on training/validation data, how often was the performance checked against test data? Technically, if task 6 to 9 was kept aside as a true test set, then it should have be evaluated once and only once at the end of all the simulations. Was that the case?"
>
> The constrained margin scores reported in Table 1 and 2 were the first evaluation of this method on these tasks; we explicitly avoided the temptation of ‘test-set leakage’. The ablation studies concerning Taylor versus Deepfool (Section B.1) and clipping (Section A.3) in the supplementary material were done after the fact. Note that the only true hyperparameter which has a large effect on results is that of the number of principal components, the selection process of which we established on the training set of tasks only, see Figure 1 in the supplementary material.
>
> We do however understand your concern that the data is now publicly available. Note that all the methods compared against in Table 2, with the exception of the techniques proposed by Natekar and Sharma, were also reported after the data was made publicly available.  We are therefore comparing these methods on equal footing.
>
> As to the results in Table 1, we have intentionally biased this comparison in favor of the other margin methods: we compare against the hidden-layer selection which performs the best across all tasks, and similarly the method of calculating standard input margins that provide the best scores. Please see sections B.1 (for input) and B.2 (for hidden) in the supplementary material for additional details on this comparison.
>
> >"The new margin definition is not motivated theoretically."
>
> Please see the global response, where we have discussed the merits of our more empirical approach.
>
> >"It seems like something sensible to do, but I am not convinced it gives us all that much insight beyond improving PGDL score.  I suspect the approach might also fail on some datasets (as evident from the performance of the proposed method on Task 5 of Table 2 (though to authors credit, the do not shy away from talking about it in the Limitations section)). It doesn't seem like a rigorous derivation of a new principle, but a bit of a hack to improve an existing approach. I presume the point of the competition was to spur new ideas and new understanding in this domain. The proposed modification seem like a minor change, and other than improving the PGDL score, I don't find any new insight in terms of learning theory. As such, this seems like an exercise in optimisation for PGDL score, and I am not sure how the proposed PCA trick relates to gaining any new understanding about generalisation principles of neural networks."
>
> Please see the global response for a discussion of this paper’s proposed contributions.
>
> We would like to add that we explicitly avoided using ‘hacks’ that could artificially increase our scores: For example, we do not combine our method with mixup-augmentation-scores, as was done for some of the complexity measures we compare against in Table 2 (DBI*LWM and PCA-Gi&Mi) even though this could potentially have improved our scores.
>
> We believe that our main finding - that the data manifold needs to be taken into account to (for the first time) relate input margins to DNN generalization -  is a relevant and useful contribution to existing margin-related studies.

---

> > ### Comment · Reviewer_Uvn7 · 2023-08-17
> > **Thanks for the response**
> >
> > Thank you for your response.  In recognition of a valiant effort to avoid 'test-set leakage' in this empirical evaluation, I can raise my score a touch, to 4, which probably doesn't change all that much.  I appreciate the authors always intended for this to be an empirical work, but that does not change the fact that it just doesn't feel very convincing, and when it comes to something like the principle of generalisation, I think benchmarking is a great beginning of a potential new direction of research, but on its own, without a bit of theoretical explanation, not very convincing.

---

### Official Review · Reviewer_hMFD · 2023-07-04

**Soundness:** 2 fair
**Presentation:** 3 good
**Contribution:** 2 fair
**Rating:** 5
**Confidence:** 4

**Summary:**

The authors propose a novel notion of constrained input margin to measure the generalization performance of neural networks. Compared to conventional input margin defined on $\mathbb{R}^n$, the proposed constrained input margin is constrained on the data manifold and estimated via g Principal Component Analysis (PCA). Extensive study has shown the decent correaltion between the constrained input margin and generalization gap of neural networks.

**Strengths:**

1. The paper is well organized and the writing is fairly clear;

2. A novel measurement of constrained margin is proposed by contraining the margin in the data manifold;

3. The empirical results show the superior correlation of constrained margin with generalziation gap of neural networks in the taks of Predicting Generalization in Deep Learning compared to other metric;

4. This work is insightful to understand the margin in the input space by taking the data manifold into consideration.

**Weaknesses:**

1. There is lack of a theoretical analysis on the contrained margins, as a theoretical analysis will make the metric more convincing to connect to model generalization;

2. As the adversarial training can reduce the conventional input margins and hurt the generalization, experiments on adversarial training are required to show the effectivness of constrained margins under different training scenarios. I will consider to increase my score if the results on adsersarial training are provided.

Overall, the manuscript is well organized and self-consistent, but more theoretical and empirical evidence are needed to verify the effectiveness of the proposed constrained margins.

**Questions:**

See weaknesses.

**Limitations:**

See weaknesses.

---

> ### Author Rebuttal · Authors · 2023-08-09
>
> We thank you for your review and suggestions. We pose the following response to the two questions raised:
>
> >"There is lack of a theoretical analysis on the contrained margins, as a theoretical analysis will make the metric more convincing to connect to model generalization."
>
> Please see the global response, where we have discussed the merits of our more empirical approach.
>
> >"As the adversarial training can reduce the conventional input margins and hurt the generalization, experiments on adversarial training are required to show the effectivness of constrained margins under different training scenarios. I will consider to increase my score if the results on adsersarial training are provided."
>
> The PGDL tasks were intended to represent DNN models under various different training scenarios, although no adversarially trained models are included. To verify whether constrained margins remain predictive under this training regime is an interesting suggestion, however, we are not able to produce such a dataset of adversarially trained models (appropriate for evaluating generalization ability) within the time constraints of the rebuttal period.
>
> To our knowledge, adversarial training *increases* standard input margins, given that the margin increases in order to fit these adversarial samples. Standard input margins will therefore inaccurately indicate better generalization, when models in fact can generalize more poorly in the standard setting (in the absence of a targeted attack) [1]. We suspect that adversarial retraining would only significantly affect the very nearby decision boundaries (depending on how drastically the model overfits), thus constrained margins would be mostly unaffected. We conclude that this is a promising avenue for future work.
>
>
> [1] Dimitris Tsipras, Shibani Santurkar, Logan Engstrom, Alexander Turner, and Aleksander
> Madry. *Robustness May Be at Odds with Accuracy*. ICLR, 2019

---

### Official Review · Reviewer_588w · 2023-07-05

**Soundness:** 2 fair
**Presentation:** 2 fair
**Contribution:** 2 fair
**Rating:** 4
**Confidence:** 4

**Summary:**

This paper deals with the relationship between input margins and generalization. Considering the decision boundary may be not inherently useful in classification, the authors develop a constrained margin based on directions extracted from input data. Principal Component Analysis (PCA) is used to identify the subspaces that are thought to contain truly useful features. In this way, the margin measurement is limited to the input space, making the method more robust without selecting specific layers as in hidden margins. A DeepFool-based algorithm is proposed to calculate the margin efficiently. Experiments show that the proposed method has a good performance on the prediction of generalization, and achieves higher score than other complexity measures for most tasks. The authors carry out further experiments to show that high utility directions are more predictive of a model’s performance that low utility directions, which aligns with the initial intuition.

####################### After Rebuttal ##############################################
I've read the authors' responses. However, I still believe that the novelty of this paper is below the average standard of NeurIPS. I'll keep my score in this period.

**Strengths:**

1. The paper is well-written and easy to follow.
2. It is interesting to measure generalization from the view of input margins, which is firstly demonstrated as the authors claim.

**Weaknesses:**

1. The approximation in Equation (3) is rough. It may be useful on small datasets and easy tasks, but can be massively computational for PCA and not indicative on more complicated datasets and occasions. The authors are expected to give examples or discuss under this setting.
2. Though the authors provide an interesting prospective, the contribution is limited as the approximation and algorithm, such as Taylor approximation and DeepFool algorithm, are actually well-known methods.
3. The claimed connection between input margins and generalization only has empirical evidence. As we all know that traditional definition of margin enjoys very strong theoretical support. Hence, I would like to see more theoretical evidence to support this conjecture.
4. The selection of the number of principal components (by the Kneedle algorithm) requires the ground-truth ranking. Thus, it remains to be discussed whether it is suitable to compare this method with others.

**Questions:**

Please address the concerns in the weakness part.

**Limitations:**

NAN

---

> ### Author Rebuttal · Authors · 2023-08-09
>
>
> We thank you for your constructive commentary and concerns. We have the following response to the four points raised:
>
> >"The approximation in Equation (3) is rough. It may be useful on small datasets and easy tasks, but can be massively computational for PCA and not indicative on more complicated datasets and occasions. The authors are expected to give examples or discuss under this setting."
>
> Equation (3) is less of an approximation than it may initially seem: we are not expressing the boundary point in terms of principal components, we are starting from an existing point and searching for a boundary point in a constrained subspace around this point.
>
> Even for larger sets (such as Imagenet) calculating principal components is tractable, and searching in the constrained margin space is in fact always less computationally expensive than searching in the hidden or input space.
>
> The PGDL challenge offers a diverse number of architectures and datasets, and is currently the standard for evaluating and comparing complexity measures.
>
> >"Though the authors provide an interesting prospective, the contribution is limited as the approximation and algorithm, such as Taylor approximation and DeepFool algorithm, are actually well-known methods."
>
> The use of a Taylor approximation or DeepFool method is not the core of the paper’s contribution. The contribution is the proposition of adapting such methods in a specific way so as to make simple input margins predictive of generalization, where they have never been before. Please see the global response for a discussion of the contribution.
>
> >"The claimed connection between input margins and generalization only has empirical evidence. As we all know that traditional definition of margin enjoys very strong theoretical support. Hence, I would like to see more theoretical evidence to support this conjecture."
>
> Please see the global response, where we have discussed the merits of our more empirical approach.
>
>
>
> >"The selection of the number of principal components (by the Kneedle algorithm) requires the ground-truth ranking. Thus, it remains to be discussed whether it is suitable to compare this method with others."
>
> The Kneedle algorithm does not require the ground truth ranking.
>
> As stated on line 227: “The number of principal components for each dataset is selected by plotting the explained variance (of the *train data*) per principal component in decreasing order on a logarithmic scale and applying the elbow method using the Kneedle algorithm from Satopaa et al [35].” No model-ranking or test data is involved in the selection of principal components.  This is a very important distinction. Please re-assess the validity and impact of our scores on the PGDL tasks with this in mind.

---

> > ### Comment · Reviewer_588w · 2023-08-16
> > **Thank you for your response**
> >
> > I want to thank the authors for their response. However, I still believe that the novelty of this paper is below the average standard of NeurIPS. I'll keep my score in this period.

---

### Official Review · Reviewer_oYf3 · 2023-07-07

**Soundness:** 3 good
**Presentation:** 3 good
**Contribution:** 3 good
**Rating:** 7
**Confidence:** 4

**Summary:**

The paper empirically studies input margins as a predictive measure for the generalization of neural networks.

**Strengths:**

The paper has a very interesting finding: the distance of the decision boundary of a neural network to the training samples is predictive of its generalization performance when such a distance is calculated with respect to a small number of principal components of the training set. This suggests new interesting directions for theoretical and empirical research to explain neural network generalization.


**Weaknesses:**

The measure presented in the paper is not predictive of generalization in a simple case of linearly separable data and a single neuron: w*x.

Consider the dataset in R^{d+1} drawn from ( 1 , 10*N(0,I_d) ) for positive labels and  ( -1 , 10*N(0,I_d) ) for negative labels.
In this case, you cannot reach the decision boundary for w=(1,0,...,0), even taking all components except the last one (corresponding to the first coordinate in the training set). So the margin should be infinity.

On the other hand, consider the dataset in R^{d+1} drawn from ( 1 , 0.1*N(0,I_d) ) for positive labels and  ( -1 , 0.1*N(0,I_d) ) for negative labels. In this case, for w=(1,0,...,0), taking the first component (or the first few) would yield a margin of 1.

Yet in both cases, w=(1,0,...,0) generalizes perfectly for both distributions.

One might argue that we should normalize each coordinate in advance. In that case, we can take the same datasets and randomly rotate them, and the problem will still hold even after normalizing each coordinate.

Also, see some remarks below in terms of presentation.

**Questions:**

Concerning the weakness mentioned above, am I missing something? If not, it would be beneficial to add this to the paper and add what should be the margin if one cannot reach the decision boundary.

157: Are the components p_i also normalized? Is it stated somewhere in the text?

216: Hidden margins are calculated when considering the output of the hidden layer as a new representation of the dataset, and the margins are calculated there? It's not entirely clear from the text.
Did you try to calculate the constrained margin over the hidden representations? How does this perform?

Can you compare Kendall's rank in Table 1 with the other measures in Table 2?

Mistake in Table 2 for Task 6, the first column is optimal.

In Figure 1, why do you choose just ten components, starting with the component of the value on the x-axis? Isn't it more natural to see the degradation in performance when you accumulate all components till the value on the x-axis? Maybe the latter components are less predictive alone, but combined with the first components, they provide better prediction? It would be worthwhile to have both comparisons.



**Limitations:**

The authors adequately addressed the limitations.

---

> ### Author Rebuttal · Authors · 2023-08-09
>
> We thank you for your detailed review and insight. The case of a single separating hyperplane is an interesting thought experiment that requires careful consideration, and we are also grateful for the other minor corrections. We pose the following responses to each of the questions:
>
> >"Concerning the weakness mentioned above, am I missing something? If not, it would be beneficial to add this to the paper and add what should be the margin if one cannot reach the decision boundary."
>
> To clarify: our setup is one in which *the same* data is used to train different models, and the margin measurements aim to identify which model will generalize better. (We are not providing an opinion using the same model but different data - the scenario sketched here.)
>
> That said, the suggested thought experiment can also be applied to our setup. Since we do z-normalize all features before calculating the principal components (see line 159), any feature with at least some variance will not become fully overshadowed by another. Since some variance is needed for a feature to be class-discriminative, an element of this feature will be included in earlier principal components as well, and the boundary of the thought experiment will remain reachable.
>
> In the linearly separable thought experiment (where a single feature is the only relevant one), moving the boundary closer or further from the midpoint between the only relevant feature will *on average* find the same constrained margin, if the two classes are balanced. The constrained margin will be an overestimate of the size of the true margin in all cases, but the comparison will remain sound.
> Note however that the closest distance to the boundary remains a proxy for the size of a much more complex space. Constrained margins should be seen as a way to characterize the generalization behavior of a DNN where more precise measurements quickly become intractable.
>
> In practice, it occasionally does happen that a boundary is not found. In such cases, the boundary point is clipped to stay within the allowed domain. See line 189. This happens in less than 0.5% of cases. We’ve now added a note to make it explicit that the clipped boundary is used when an actual boundary is not found.
>
> >"157: Are the components p_i also normalized? Is it stated somewhere in the text?"
>
> Yes. The eigenvectors are normalized to unit length. A note to this effect has been added at line 161.
>
> >"216: Hidden margins are calculated when considering the output of the hidden layer as a new representation of the dataset, and the margins are calculated there? It's not entirely clear from the text."
>
> Yes, the output (post activation function) of that hidden layer is considered to be the representation at which the margin is then calculated, as originally defined in [1, 2].  We now point this out in Section 2.1.
>
> >"Did you try to calculate the constrained margin over the hidden representations? How does this perform?"
>
> No, we did not try this, for two reasons:
>
> (i) The concerns with using hidden margins relate more to normalization - how to normalize across different architectures or layers -  and less to measuring margins without leaving the data manifold, the issue we address here.  (See line 136.)
>
> (ii) This would require calculating the principal components of the representations (which are up to 524k dimensions, compared to the 3k of the input space) for each layer of each model individually, which becomes prohibitively expensive, computationally and memory-wise.
>
> >"Can you compare Kendall's rank in Table 1 with the other measures in Table 2?"
>
> Table 4 in the supplementary material provides the CMI scores corresponding to Kendall's rank scores shown in Table 1 of the main paper. A direct comparison can, therefore, be made between Table 2 in the main paper and Table 4 in the supplementary. We have now combined these into a single table in the supplementary material for an easier comparison.
>
> >"Mistake in Table 2 for Task 6, the first column is optimal."
>
> Thank you!  We have corrected the misplaced bold font.
>
> >"In Figure 1, why do you choose just ten components, starting with the component of the value on the x-axis? Isn't it more natural to see the degradation in performance when you accumulate all components till the value on the x-axis? Maybe the latter components are less predictive alone, but combined with the first components, they provide better prediction? It would be worthwhile to have both comparisons."
>
> We investigated 10-dimensional subspaces of decreasing utility to determine whether ‘useful’ directions are indeed required, or whether constraining the margin search in other ways would give a similar effect. By taking different but same-sized subspaces to search in, we can directly compare the effect of using higher and lower utility subspaces: we measure a significant drop in predictive performance when using lower-utility subspaces.
>
> That said, the alternative is a valuable suggestion and we have implemented it by calculating the DeepFool constrained margins using an increasing number of principal components (Figure 1 in response PDF). One would expect that increasing the number of components would result in a decrease in predictive performance until it approaches that of standard input margins. This is also what we observe. We have included the new figure in the supplementary material. This serves to further make our main point: standard input margins are essentially useless on their own, and that the search needs to be constrained to an appropriate subspace for these margins to be predictive of generalization.
>
> [1] Gamaleldin Elsayed, Dilip Krishnan, Hossein Mobahi, Kevin Regan, and Samy Bengio. *Large margin deep networks for classification*. NeurIPS, 2018
>
> [2] Yiding Jiang, Dilip Krishnan, Hossein Mobahi, and Samy Bengio.
> *Predicting the generalization gap in deep networks with margin distributions*. ICLR, 2018

---

> > ### Comment · Reviewer_oYf3 · 2023-08-14
> >
> >
> > Thank you for your response!
> >
> > Since I'm mainly interested in the example I described, I'll start by addressing your comments about it.
> >
> > **"To clarify: our setup is one in which the same data is used to train different models, and the margin measurements aim to identify which model will generalize better. (We are not providing an opinion using the same model but different data - the scenario sketched here.)"**
> >
> > Thanks for clarifying this for me!
> >
> > Nevertheless, shouldn't a good generalization measure also correlate well across datasets? Do you have a way of examining and presenting that?
> >
> > **"That said, the suggested thought experiment can also be applied to our setup. Since we do z-normalize all features before calculating the principal components (see line 159), any feature with at least some variance will not become fully overshadowed by another. Since some variance is needed for a feature to be class-discriminative, an element of this feature will be included in earlier principal components as well, and the boundary of the thought experiment will remain reachable."**
> >
> > I figured you would answer as above. This is why I continued as follows in my original review:
> >
> > "One might argue that we should normalize each coordinate in advance. In that case, we can take the same datasets and randomly rotate them, and the problem will still hold even after normalizing each coordinate."
> >
> > I think that the problem I raised still holds. Am I missing something?

---

> > > ### Author Response · Authors · 2023-08-16
> > >
> > > Thank you for your follow-up!
> > >
> > > >"Nevertheless, shouldn't a good generalization measure also correlate well across datasets? Do you have a way of examining and presenting that?"
> > >
> > > We are assuming that, by “also correlate well across datasets”, you mean that the measure should also be predictive of generalization for models trained on different datasets. Please clarify if we misunderstood!
> > >
> > > Comparing the generalization of models that are trained on different datasets is an interesting avenue of exploration, but not the one explored in the context of the PGDL dataset. The complexity measures considered here do not speak to the *quality* of the training data, or attempt to answer which data-qualities result in good generalization. Rather, we attempt to answer a different question: If two models fit the same training data equally well, but differ greatly in their generalization performance, which model characteristics distinguish one model from another? In doing so we are trying to uncover which elements lead to improved generalization. In this context, mean constrained margin is a new and reliable indicator.
> > >
> > > >"I figured you would answer as above. This is why I continued as follows in my original review:
> > > "One might argue that we should normalize each coordinate in advance. In that case, we can take the same datasets and randomly rotate them, and the problem will still hold even after normalizing each coordinate."
> > > I think that the problem I raised still holds. Am I missing something?"
> > >
> > > Right, sorry! If the normalized data is rotated such that the principal component is parallel to the decision boundary, it would imply that the model in this thought experiment no longer separates the data accurately. Our underlying assumption is that the model performs well on the training data. Note that for the PGDL tasks, all models are trained to >99% train accuracy, so this assumption clearly holds. The only edge case we have encountered is when the decision boundaries are created so far away that they are outside the feature range of the dataset, which we explained in the previous response.

---

> > > > ### Comment · Reviewer_oYf3 · 2023-08-17
> > > >
> > > > >We are assuming that, by “also correlate well across datasets”, you mean that the measure should also be predictive of generalization for models trained on different datasets. Please clarify if we misunderstood!
> > > >
> > > > Yes, this is what I meant.
> > > >
> > > > >Right, sorry! If the normalized data is rotated such that the principal component is parallel to the decision boundary, it would imply that the model in this thought experiment no longer separates the data accurately.
> > > >
> > > > What you described was not my intention. Let me give an exact description:
> > > >
> > > > Fix $d>>1$.
> > > >
> > > > * You draw a large sample size n: n/2 from ( 1 , 0.1N(0,I_d) ) for positive labels and n/2 from ( -1 ,0.1N(0,I_d) ) for negative labels. Let's call this dataset $X1_{(d+1)\times n}$ with the labels $Y1_{1\times n}$ as above fit it with a weight vector $w$ that optimally separates the data so $w=(1,0,0,0,...,0)$.  Take now a random orthogonal matrix $M$. Define a new dataset $Z1=M \cdot X1$  with the original labels  $Y$. It is the same dataset geometrically since $M$ is orthogonal. Fit the new dataset with a new weight vector $u$ that optimally separates the dataset so $u=w\cdot M^T $.
> > > >
> > > > * Repeat the process above with a large sample size n: n/2 from $( 1 , 10N(0,I_d) )$ for positive labels and n/2 from $( -1 ,10N(0,I_d) )$ for negative labels. Let's call this dataset $X2_{(d+1)\times n}$ with the labels $Y2_{1\times n}$ as above fit it with a weight vector $w$ that optimally separates the data so $w=(1,0,0,0,...,0)$.  Take now a random orthogonal matrix $M$. Define a new dataset $Z2=M \cdot X2$   with the original labels  $Y2$. It is the same dataset geometrically since $M$ is orthogonal. Fit the new dataset with a new weight vector $u$ that optimally separates the dataset so $u=w\cdot M^T $.
> > > >
> > > > $u$ for datasets $Z1,Y1$ and $Z2,Y2$ generalizes perfectly, yet **I think** your measure would predict good accuracy for the first and bad for the second, assuming you are only considering a small number of principal components, say 10. This would hold even if you normalize the coordinates of $Z1$ and $Z2$ in advance.
> > > >
> > > > Makes sense?

---

> > > > > ### Author Response · Authors · 2023-08-18
> > > > >
> > > > > Thank you for the clarification!
> > > > >
> > > > > We now understand what you mean! You are correct: z-normalizing the data *after* rotation would not solve the variance discrepancy between the two datasets in question. This perhaps emphasises that our assumption is that the directions of high variance correspond to those of high utility, which is not the case in this thought experiment. Do you think it would be beneficial to include this thought experiment in the manuscript?

---

> > > > > > ### Comment · Reviewer_oYf3 · 2023-08-21
> > > > > >
> > > > > > Yes, I think it would be beneficial to include this thought experiment in the manuscript. In the sense that a reader would understand that the measure might still fail in some situations.
> > > > > >
> > > > > >  Although for a casual reader, it might seem like a weakness of your generalization measure, I believe it's a strength as it highlights some inherent properties of natural distributions (I personally do not believe that a global generalization measure exists). Somehow for natural distributions, as you said:  "directions of high variance correspond to those of high utility"  an observation that I find quite intriguing!
> > > > > >
> > > > > > Other than that,  you addressed my questions fully, and I'm very happy to recommend your paper for publication since it brings a novel, simple, concise, and interesting empirical finding! (although I'm a minority among the other reviewers)

---

> > > > > > > ### Author Response · Authors · 2023-08-21
> > > > > > >
> > > > > > > We thank you for your insight and assessment.
> > > > > > > We have included a description of this thought experiment in Section 3.

---

### Author Rebuttal · Authors · 2023-08-09

We would like to thank all reviewers for their commentary; we believe this has been a constructive exercise in improving the manuscript. We have addressed each reviewer’s comments individually. In addition, we would like to make two points:


1. There is a strong line of research that investigates the link between classification margins and generalization in DNNs from an empirical perspective [1, 2, 3, 4]. This is also the case for work that follows after the PGDL-challenge: for example, we find no direct theoretical contributions in [5, 6]. Rather, a theoretical intuition is empirically confirmed, as we do here as well. Our work therefore *is* an empirical study, and well aligned with an existing body of research. In contrast, work such as that of Chuang et al. [7] that do provide a theoretical analysis, was mainly concerned with *output* margins and normalized variants thereof, which are more commonly considered from the theoretical perspective [8, 9].

2. We believe our contribution goes beyond improving performance on the PGDL tasks.
We make the first clear link between input margin and generalization in DNN-based classifiers. This should be clear from Table 1: in most cases the standard input margin is not useful for generalization prediction. Our finding that input margins can become useful if the data manifold is taken into account is a new insight. This is not a trivial finding: hidden margins require normalization when comparing across architectures: how to do this in the general case remains unresolved. At the same time, the input space remains exactly the same, irrespective of model architecture. Techniques based on the input space, therefore, require no per-layer normalization or hidden layer selection, and are not burdened with the per-task finetuning this can require.
In addition, this opens up a number of new avenues for research: will a more complex data manifold modeling technique produce different results? Can the constrained margins be used to improve generalization? For example, both [2] and [10] have developed techniques to maximize margins during training but these have not yet resulted in improved generalization. If constrained margins are more correlated with generalization prediction than earlier margin-based measures, will maximizing constrained margins produce a different effect?

In conclusion, we believe that our main finding - that the data-manifold needs to be taken into account to relate input margins to DNN generalization - has been demonstrated empirically, and is a relevant and useful contribution to the DNN generalization literature.


Content of PDF: Please see the final point in the response to reviewer oYf3

[1] Yiding Jiang, Dilip Krishnan, Hossein Mobahi, and Samy Bengio.
*Predicting the generalization gap in deep networks with margin distributions*. ICLR, 2018

[2] Gamaleldin Elsayed, Dilip Krishnan, Hossein Mobahi, Kevin Regan, and Samy Bengio. *Large margin deep networks for classification*. NeurIPS, 2018

[3] David Stutz, Matthias Hein, and Bernt Schiele.
*Disentangling Adversarial Robustness and Generalization*. CVPR, 2019

[4] Parth Natekar and Manik Sharma.
*Representation based complexity measures for predicting generalization in deep learning*. NeurIPS 2020, PGDL challenge.

[5] Yair Schiff, Brian Quanz, Payel Das, and Pin-Yu Chen.
*Predicting Deep Neural Network Generalization with Perturbation Response Curves*. NeurIPS, 2021

[6] Yi Zhang, Arushi Gupta, Nikunj Saunshi, and Sanjeev Arora. *On Predicting Generalization using GANs*. ICLR, 2022

[7] Ching-Yao Chuang, Youssef Mroueh, Kristjan Greenewald, Antonio Torralba, and Stefanie Jegelka. *Measuring generalization with optimal transport*. NeurIPS, 2021

[8] Peter L Bartlett, Dylan J Foster, and Matus J Telgarsky. *Spectrally-normalized margin bounds for neural networks*. NeurIPS, 2017

[9] Behnam Neyshabur, Srinadh Bhojanapalli, and Nathan Srebro. *A pac-bayesian approach to spectrally-normalized margin bounds for neural networks*. ICLR, 2018

[10] Yuancheng Xu, Yanchao Sun, Micah Goldblum, Tom Goldstein, and Furong Huang. *Exploring and Exploiting Decision Boundary Dynamics for Adversarial Robustness*. ICLR, 2023.

---

### Decision · Program_Chairs · 2023-09-21

**Decision:**

Reject

**Comment:**

This paper had widely varying scores from reviewers after rebuttal. The key schism was the need for "theoretical justification", which 2 of the reviewers felt was missing. Personally, I do not believe that it is important to have any theoretical justification if the idea is interesting and intuitive, and backed by empirical evidence, which are all true in this case. However, given the overall low enthusiasm of one of the positive reviewers, and concerns raised about some of the empirical results, I sadly have to reject the paper. I feel it is close to the bar for acceptance, and encourage the authors to take into account the feedback from all reviewers, and resubmit to another venue.